# Phase Angle as an Indicator of Sarcopenia, Malnutrition, and Cachexia in Inpatients with Cardiovascular Diseases

**DOI:** 10.3390/jcm9082554

**Published:** 2020-08-06

**Authors:** Suguru Hirose, Toshiaki Nakajima, Naohiro Nozawa, Satoshi Katayanagi, Hayato Ishizaka, Yuta Mizushima, Kazuhisa Matsumoto, Kaori Nishikawa, Yohei Toyama, Reiko Takahashi, Tomoe Arakawa, Tomohiro Yasuda, Akiko Haruyama, Hiroko Yazawa, Suomi Yamaguchi, Shigeru Toyoda, Ikuko Shibasaki, Takashi Mizushima, Hirotsugu Fukuda, Teruo Inoue

**Affiliations:** 1Department of Cardiovascular Medicine, School of Medicine, Dokkyo Medical University, Shimotsuga-gun, Tochigi 321-0293, Japan; suguru8@dokkyomed.ac.jp (S.H.); hal@dokkyomed.ac.jp (A.H.); hyzw@dokkyomed.ac.jp (H.Y.); suomi@dokkyomed.ac.jp (S.Y.); s-toyoda@dokkyomed.ac.jp (S.T.); inouet@dokkyomed.ac.jp (T.I.); 2Department of Rehabilitation, Dokkyo Medical University Hospital, Shimotsuga-gun, Tochigi 321-0293, Japan; n-nozawa@dokkyomed.ac.jp (N.N.); kata-s@dokkyomed.ac.jp (S.K.); i-hayato@dokkyomed.ac.jp (H.I.); yu-mizu@dokkyomed.ac.jp (Y.M.); m-kazu@dokkyomed.ac.jp (K.M.); knishi@dokkyomed.ac.jp (K.N.); y-toyama@dokkyomed.ac.jp (Y.T.); reiko-m@dokkyomed.ac.jp (R.T.); tomoe-a@dokkyomed.ac.jp (T.A.); mizusima@dokkyomed.ac.jp (T.M.); 3School of Nursing, Seirei Christopher University, Hamamatsu, Shizuoka 433-8558, Japan; tomohiro-y@seirei.ac.jp; 4Department of Cardiovascular Surgery, School of Medicine, Dokkyo Medical University, Shimotsuga-gun, Tochigi 321-0293, Japan; sibasaki@dokkyomed.ac.jp (I.S.); fukuda-h@dokkyomed.ac.jp (H.F.)

**Keywords:** phase angle, skeletal muscle mass index, cardiovascular disease, nutrition, CONUT score, sarcopenia, cachexia

## Abstract

Malnutrition is associated with sarcopenia, cachexia, and prognosis. We investigated the usefulness of phase angle (PhA) as a marker of sarcopenia, cachexia, and malnutrition in 412 hospitalized patients with cardiovascular disease. We analyzed body composition with bioelectrical impedance analysis, and nutritional status such as controlling nutritional status (CONUT) score. Both skeletal muscle mass index (SMI) and PhA correlated with age, grip strength and knee extension strength (*p* < 0.0001) in both sexes. The SMI value correlated with CONUT score, Hb, and Alb in males. Phase angle also correlated with CONUT score, Hb, and Alb in males, and more strongly associated with these nutritional aspects. In females, PhA was correlated with Hb and Alb (*p* < 0.001). In both sexes, sarcopenia incidence was 31.6% and 32.4%; PhA cut-off in patients with sarcopenia was 4.55° and 4.25°; and cachexia incidence was 11.5% and 14.1%, respectively. The PhA cut-off in males with cachexia was 4.15°. Multivariate regression analysis showed that grip strength and brain natriuretic peptide (BNP) were independent determinants of SMI, whereas grip strength, BNP, and Hb were independent determinants of PhA. Thus, PhA appears to be a useful marker for sarcopenia, malnutrition, and cachexia in hospitalized patients with cardiovascular disease.

## 1. Introduction

Malnutrition and increased nutritional risk are frequently observed among hospitalized patients and cardiovascular disease (CVD) patients [1,2]. At-risk patients have prolonged hospital stays, increased rates of hospitalization and readmission, increased prevalence of treatment-related complications, and higher mortality [3,4]. The pathophysiology of increased nutritional risk includes changes in appetite and dietary intake and development of catabolism, followed by a decrease in physical function and muscle mass. In addition, as life expectancy increases, sarcopenia, the skeletal muscle loss, and diminished physical function (grip strength, walking speed) common in the elderly population [5], is becoming a major health issue [6]. It is frequently associated with CVD [7] including heart failure (HF) and chronic kidney disease (CKD) [5,8]. Thus, it is generally accepted that increased nutritional risk, malnutrition, and sarcopenia are predictors of survival in patients with CVD, and they increase the risk of complications and mortality [6,9]. Especially, chronic HF has a prevalence of 1% in the general population [10]. Due to the fact of its symptoms, it has profound effects on quality of life, and it is a common reason for hospitalization and greater overall mortality [10]. Patients with HF have a high prevalence of increased nutritional risk [11]. Sarcopenia in chronic HF may ultimately lead to tissue wasting and cardiac cachexia which is associated with an extremely poor prognosis [12,13,14].

Various anthropometric methods are widely used and could be considered cornerstones for the assessment of nutritional status. Bioelectrical impedance analysis (BIA) is a promising nutritional assessment tool that incorporates both functional and morphological evaluation. It enables measurement of the differences in the electrical resistance of various tissues (e.g., fat, muscle, and bone) through application of a weak current to the body. In particular, the clinically established bioelectrical impedance parameter is the phase angle (PhA), defined as the ratio of resistance (R, intracellular and extracellular resistance) to reactance (Xc, cell membrane-specific resistance) [15]. It reflects cellular vitality and integrity, where normal values indicate preserved cellular activity [16,17,18,19]. It may become an important tool in assessing nutritional status in any situation, being superior to anthropometric and biochemical methods [20,21] and has also been studied as a highly predictive index of impaired clinical outcomes and mortality for various diseases including HF [19,22]. However, a limited number of studies evaluated the clinical applications of the PhA such as a more accurate identification of malnourished patients with CVD.

Therefore, we investigated the clinical usefulness of the PhA as a marker of sarcopenia, malnutrition, and cachexia in hospitalized patients with CVD.

## 2. Materials and Methods

### 2.1. Participants

A total of 412 patients who underwent cardiac rehabilitation on admission due to the CVD were included in this study. Their baseline characteristics are summarized in Table 1. Two hundred and seventy-seven patients were males (67%) and 135 patients were females (33%). The mean age and body mass index (BMI) of the males were 67.7 ± 12.5 years and 23.6 ± 3.8 kg/m^2^, respectively. The mean age and BMI of the females were 74.6 ± 11.4 years and 22.1 ± 4.3 kg/m^2^, respectively. We excluded the following types of patients: (1) patients with cerebrovascular disease and those undergoing arthroscopic joint surgery; (2) patients with chronic diseases such as severe orthopedic disorders, malignancies, or cognitive dysfunction; (3) patients with pacemaker implantation and a contraindication for BIA methods. Fifty-nine patients underwent coronary artery bypass grafting (CABG), 62 had valve replacement or repair, 50 had aortic surgery including endovascular aneurysm repair (EVAR) and artificial blood vessel replacement. Eight had arteriosclerosis obliterans (ASO) and 26 had transcatheter aortic valve implantations (TAVI). One hundred and seventeen patients had congestive heart failure (CHF) and 95 patients had ischemic heart disease (IHD), including angina pectoris and myocardial infarction. We assessed the co-incidence of conventional risk factors as shown in Table 1. The study protocol conformed to the ethical guidelines of the Declaration of Helsinki as reflected in a priori approval by the institutional human research committee. The proposal was approved by the Regional Ethics Committee of Dokkyo Medical University Hospital.

All participants underwent complete laboratory chemistry and hematologic evaluation. Fasting venous blood samples were collected in tubes containing EDTA sodium (1 mg/mL) and in polystyrene tubes without an anticoagulant. Plasma was immediately separated by centrifugation at 3000 rpm at 4 °C for 10 min, and serum was collected by centrifugation at 1000 rpm at room temperature for 10 min. Brain natriuretic peptide (BNP) and estimated glomerular filtration rate (eGFR) were measured. Blood hemoglobin (Hb), albumin (Alb), and total cholesterol (TChol) levels were analyzed with routine chemical methods in the Dokkyo Medical University Hospital clinical laboratory.

### 2.2. Short Physical Performance Battery

Participants completed the Short Physical Performance Battery (SPPB) according to the National Institute on Aging protocol. The tests were performed in the following sequence: (a) standing balance tests, (b) gait test (4 m), and (c) chair stand test (5 repetitions). The standing balance portion requires participants to maintain the body for 10 s each, which includes stances with their feet placed side by side, semi-tandem, and in tandem. The scores range from 0–4 (maximum performance). The gait test measures the time required to walk 4 m at a typical pace. The chair stand test requires participants to rise from a steel chair (0.40 m in height and 0.30 m in depth) with their arms across their chest, five times. Categorical scores (range: 0–4) for both the gait and the chair stand tests were based on timed quartiles established previously in a large population. Individuals who were unable to complete either the 4 m gait task or the 5 repetitions in the chair stand test received a score of 0. The sum of the three components comprised the final SPPB score with a possible range from 0–12. A score of 12 indicated the highest degree of lower extremity function [21].

### 2.3. Measurement of Gait Speed, Grip Strength, and Voluntary Isometric Contraction

Maximum voluntary isometric contraction (MVIC) of the hand grip was determined with a factory-calibrated hand dynamometer (TKK 5401, TAKEI Scientific Instruments Co., Ltd., Tokyo, Japan). Each subject underwent two trials, and the highest value of the two trials was used for analysis. The MVIC of the knee extensors was determined with a digital handheld dynamometer (μTas MT-1, ANIMA Co., Ltd., Tokyo, Japan) as described previously [23,24]. Each subject performed two trials with an interval of at least 2 min between trials, and the highest score was used for analysis.

### 2.4. Measurements with the Bioelectrical Impedance Analyzer (BIA)

A multi-frequency bioelectrical impedance analyzer (BIA), InBody S10 Biospace device (Biospace Co., Ltd., Seoul, Korea/Model JMW140) was used according to the manufacturer’s guidelines as described in detail previously [23,24]. Thirty impedance measurements were obtained using 6 different frequencies (1, 5, 50, 250, 500, and 1000 kHz) at the following 5 segments of the body: right and left arms, trunk, and right and left legs. The measurements were carried out while the subjects rested quietly in the supine position, with their elbows extended and relaxed along their trunk. Bioelectrical impedance analyzer-derived body components, such as body fat volume, fat-free mass (FFM), FFMI index (FFMI), lean mass index (LMI, lean mass/height^2^), skeletal muscle volume, body cell mass (BCM), % body fat, extracellular water (ECW), total body water (TBW), and PhA values were recorded. The value of ECW/TBW was calculated based on the ratio of ECW and TBW results. The PhA was calculated with resistance (R) and reactance (Xc; measured at 50 kHz) with the following equation:PhA (°) = arctangent (Xc/R) × (180/π)(1)

The skeletal muscle mass index (SMI; appendicular skeletal muscle mass/height^2^, kg/m^2^) was measured as the sum of lean soft tissue of the two upper limbs and the two lower limbs. In this study, sarcopenia was defined according to the Asian Working Group for Sarcopenia (AWGS) [5] criteria (age, ≥65 years; hand grip strength, <26 kgf for males and <18 kgf for females; gait speed, ≤0.8 m/s; SMI, <7.0 kg/m^2^ for males and <5.7 kg/m^2^ for females). Cachexia has been defined by Evans et al. [25] as a loss of lean tissue mass, involving a weight loss greater than 5% of body weight in 12 months or less in the presence of chronic illness or as BMI lower than 20 kg/m^2^. In addition, usually three of the following five criteria are required: decreased muscle strength, fatigue, anorexia, low fat-free mass index (FFMI), increase of inflammation markers such as C-reactive protein or interleukin (IL)-6 as well as anemia or low serum albumin (CRP > 5.0 mg/L, IL-6 > 4.0 pg/mL, Hb < 12 g/dL, Alb < 3.2 g/dL). In the present study, cachexia was determined to meet BMI < 20 kg/m^2^ and FFMI (≤17.4 kg/m^2^ for males and ≤15 kg/m^2^ for females) [26] and at least two of the following biochemical criteria (Hb < 12 g/dL, CRP < 5 mg/dL, Alb < 3.2 g/dL).

### 2.5. Controlling Nutritional Status (CONUT) Score

The CONUT score, which is calculated based on the serum Alb concentration (range: 0–6), the total peripheral lymphocyte count (range: 0–3), and TChol concentration (range: 0–3), was developed as a screening tool for early detection of poor nutritional status. The sum of the three components comprised the final CONUT score with a possible range from 0–12. A score of zero indicated the poorest nutritional status [27].

### 2.6. Measurement of Muscle Thickness by Ultrasound

Ultrasound evaluation of quadriceps muscle thickness was measured at the midpoint of the thigh length with a real-time linear electronic scanner with a 10.0 MHz scanning head and Ultrasound Probe (L4–12t-RS Probe, GE Healthcare Japan) and LOGIQ e ultrasound (GE Healthcare Japan) as previously described [23,24]. The scanning head was coated with a water-soluble transmission gel to provide acoustic contact without depressing the dermal surface. The subcutaneous adipose tissue–muscle interface and the muscle–bone interface were identified from the ultrasonic image. The perpendicular distance from the adipose tissue–muscle interface to the muscle–bone interface was considered to represent the quadriceps muscle thickness. The anterior thigh muscle thickness (MTH) was measured in the supine position; the measurement was performed twice at each side of the thigh, and the average value was adopted.

### 2.7. Statistical Analysis

Data are presented as the mean ± SD. The comparison of means among groups was carried out with a Mann–Whitney *U*-test or Student *t*-test. When the data were not normally distributed, non-parametric statistical analysis with the Kolmogorov–Smirnov test was performed. Associations among parameters were evaluated with Pearson or Spearman correlation coefficients. Receiver operating characteristic (ROC) curves were plotted to identify an optimal PhA cut-off for detecting sarcopenia or cachexia. With or without sarcopenia or cachexia as dependent factors, the sensitivity, specificity, and false positive rate (1-specificity) of the phase angle were calculated to obtain the ROC curve. At this time, the Youden index (sensitivity + specificity − 1) was calculated from the obtained sensitivity and specificity, and the point at the maximal value was taken as the optimum cut-off value. Multivariate linear regression analysis with the PhA as the dependent variable was performed to identify the independent factors (clinical laboratory data or physical data) that influenced it. Age and BMI were covariates. When the independent data were not normally distributed, they were logarithmically transformed to achieve a normal distribution. All analyses were performed with SPSS version 24 (IBM Corp., New York, NY, USA) for Windows. A *p*-value less than 0.05 was regarded as significant.

## 3. Results

### 3.1. Physical Characteristics and Clinical Data

Age and % body fat were greater in females than in males (*p* < 0.0001), but standing height, body weight, and BMI were greater in males than in females (*p* < 0.0001) (Table 1). Functional measurements and morphological assessments determined by BIA methods and muscle echocardiographic findings are also shown in Table 1. Hand grip strength, knee extension strength, anterior MTH, PhA, LMI, SMI, gait speed, and SPPB (total) were higher in males than in females (*p* < 0.0001). The mean PhA and SMI were 4.79 ± 1.02° and 6.99 ± 1.06 kg/m^2^, respectively, in males, and 3.99 ± 0.89° and 5.20 ± 0.96 kg/m^2^, respectively, in females. The anterior MTH was also higher in males (2.55 ± 0.78 cm) than in females (2.04 ± 0.66 cm; *p* < 0.0001). Systolic blood pressure (BP), diastolic BP, BNP, Alb, CONUT score, and eGFR were not significantly different between males and females. The Hb level in males was significantly higher than that in females (12.4 ± 2.2 g/dL version 11.5 ± 1.5 g/dL, respectively; *p* < 0.0001). The TChol value in males was significantly lower than that in females (160.9 ± 40.6 mg/dL version 184.8 ± 47.1 mg/dL, respectively; *p* < 0.0001).

### 3.2. Correlations among Various Parameters and SMI, PhA, and Anterior MTH

The correlations among the SMI, PhA, and anterior thigh MTH and the clinical data are shown in Table 2. The SMI, PhA, and anterior thigh MTH correlated negatively with age but positively with BMI in both sexes. As shown in Figure 1A, SMI correlated positively with both PhA (Figure 1Aa) and anterior thigh MTH (Figure 1Ab) in the total population. The SMI correlated positively with PhA and anterior thigh MTH in both sexes (Table 2). The PhA also correlated with anterior thigh MTH in both sexes. The SMI, PhA, and anterior thigh MTH correlated well with muscle strength (hand grip and knee extension strength) and SPPB (total) in both sexes. Gait speed correlated positively with the SMI (*r* = 0.2402, *p* < 0.001), PhA (*r* = 0.4112, *p* <0.0001), and anterior thigh MTH (*r* = 0.3070, *p* < 0.0001) in males. On the other hand, gait speed did not correlate significantly with the SMI and PhA but did with the anterior thigh MTH (*r* = 0.2932, *p* < 0.001) in females.

The CONUT score correlated negatively with the PhA (*r* = −0.4190, *p* < 0.0001), SMI (*r* = −0.2088, *p* < 0.05), and anterior thigh MTH (*r* = −0.2753, *p* < 0.001) in males (Table 2). In contrast, the CONUT score did not correlate significantly with the PhA and SMI in females. The Hb level correlated positively with the PhA in both sexes (*r* = 0.4668, *p* < 0.0001 for males, *r* = 0.2066, *p* < 0.05 for females). However, the Hb level was correlated with SMI (*r* = −0.2385, *p* < 0.001) and anterior thigh MTH (*r* = −0.4067, *p* < 0.0001) in males, but not females. Similarly, the Alb level correlated positively with the PhA in both sexes (*r* = 0.4055, *p* < 0.0001 for males, *r* = 0.3115, *p* < 0.001 for females). The Alb level correlated weakly with SMI (*r* = 0.1424, *p* < 0.05) in males but not females and positively with anterior thigh MTH in both sexes (*r* = 0.2467, *p* < 0.0001 for males, *r* = 0.4840, *p* < 0.0001 for females). Figure 1B shows the relationships between Alb and SMI/PhA in all patients. The Alb level correlated positively with SMI (*r* = 0.1424, *p* < 0.05, Figure 1Ba) and the PhA (*r* = 0.4054, *p* < 0.0001, Figure 1Bb).

As shown in Table 2, the eGFR correlated positively with the SMI (*r* = 0.1679, *p* < 0.01), PhA (*r* = 0.4183, *p* < 0.0001), and anterior thigh MTH (*r* = 0.2360, *p* < 0.001) in males. It also correlated positively with the PhA (*r* = 0.2549, *p* < 0.001) but not the SMI and anterior thigh MTH. The BNP level correlated negatively with SMI (*r* = −0.2397, *p* < 0.01), PhA (*r* = −0.3527, *p* < 0.0001), and anterior thigh MTH (*r* = −0.3858, *p* < 0.0001) in males (Table 2). It also correlated negatively with the PhA (*r* = −0.3268, *p* < 0.0001) but not the SMI and anterior thigh MTH in females.

### 3.3. Multivariate Regression Analysis of the PhA/SMI and Clinical Parameters

Results of the linear regression analysis with the PhA or SMI as the dependent variable and clinical data (hand grip strength, knee extension, BNP, eGFR, CRP, Hb, and Alb) as independent variables in both males and females are shown in Table 3. Univariate regression analysis showed that both hand grip strength (β = 0.378, *p* = 0.000 for males, β = 0.453, *p* = 0.001 for females) and Hb level (β = 0.291, *p* = 0.000 for males, β = 0.230, *p* = 0.038 for females) were independent predictors of the PhA in both sexes, while BNP (β = −0.239, *p* = 0.004) and CRP levels (β = 0.176, *p* = 0.035) were independent predictors of the PhA in males. On the other hand, univariate regression analysis showed that hand grip strength (β = 0.554, *p* = 0.000 for males, β = 0.593, *p* = 0.000 for females) was an independent predictor of the SMI in both sexes, while BNP level (β = −0.311, *p* = 0.001) was an independent predictor of the SMI in males.

Multivariate regression analysis showed that hand grip strength (β = 0.316, *p* = 0.000 for males, β = 0.320, *p* = 0.000 for females) was an independent predictor of the PhA in both sexes, while BNP (β = −0.206, *p* = 0.012) and Hb levels (β = 0.227, *p* = 0.005) were independent predictors of the PhA in males, even after adjusting for BMI and age. Alternatively, multivariate regression analysis showed that hand grip strength (β = 0.376, *p* = 0.000 for males, β = 0.266, *p* = 0.024 for females) was an independent predictor of the SMI in both sexes, and BNP level (β = −0.198, *p* = 0.005) was an independent predictor of the SMI, after adjusting for BMI and age.

### 3.4. Relationships Between Sarcopenia and Various Clinical Parameters

Sarcopenia was identified in 31.6% of 234 males and 32.5% of 114 females based on the sarcopenia criteria. Patients with sarcopenia were significantly older (both sexes) and had higher BNP levels (males), compared with those without sarcopenia (Table 4). On the other hand, patients with sarcopenia had significant lower gait speed, hand grip strength, knee extension strength, SMI, LMI, SPPB (total), PhA, and anterior thigh TMH than patients without sarcopenia. The mean SMI in patients with sarcopenia (5.92 ± 0.57 kg/m^2^ in males and 4.58 ± 0.61 kg/m^2^ in females) was lower than that in patients without sarcopenia (7.54 ± 0.84 kg/m^2^ in males and 5.64 ± 0.85 kg/m^2^ in females). The mean PhA in patients with sarcopenia (4.05 ± 0.79° in males and 3.62 ± 0.69° in females) was lower than that in patients without sarcopenia (5.19 ± 0.87° in males and 4.30 ± 0.88° in females). The mean anterior thigh TMH in patients with sarcopenia (1.96 ± 0.64 cm in males and 1.75 ± 0.57 cm in females) was lower than that in patients without sarcopenia (2.91 ± 0.67 cm in males and 2.28 ± 0.64 cm in females). Cachexia was identified in 11.5% males and 14.1% females based on the cachexia criteria of the present study.

The ROC curves were plotted to identify the optimal PhA cut-offs for detecting sarcopenia or cachexia in both males and females. To construct the ROC curves, different PhA cut-off values were used to predict sarcopenia (Figure 2A) and cachexia (Figure 2B) with true positives plotted on the vertical axis (sensitivity) and false-positives (1-specificity) plotted on the horizontal axis. The area under the curve (AUC) for PhA to detect sarcopenia was 82.1% in males and 77.7% in females. Sensitivity and specificity were 76.0% and 74.0%, respectively, in males and 61.4% and 86.8%, respectively, in females. The optimal PhA cut-off was 4.55° for males and 4.25° for females, as shown in Figure 2A. In addition, the AUC for the PhA to detect cachexia was 83.1% in males and 67.8% in females. Sensitivity and specificity were 79.9% and 74.2% in males, and the optimal PhA cut-off was 4.12° for males, as shown in Figure 2B. In contrast, the sensitivity of AUC curve in females was low (39.1%).

## 4. Discussion

The major findings of the present study were as follows: (1) In a total of 412 hospitalized patients with CVD, both the SMI and PhA correlated negatively with age, but positively with BMI, hand grip strength, knee extension strength, and SPPB score in both sexes. (2) The SMI and PhA were significantly associated with CONUT score, Hb, and Alb level in men, but the PhA was more strongly associated with them. In females, the PhA, but not SMI, was correlated with Hb and Alb levels. (3) Sarcopenia was found in 31.6% of men and 32.4% of women. The PhA cut-off obtained from the ROC curve in patients with sarcopenia was 4.55° for men and 4.25° for females. The PhA cut-off obtained from the ROC curve for men with cachexia was 4.15°. (4) Multivariate regression analysis showed that hand grip strength and BNP level were independent determinants of SMI, whereas grip strength, BNP, and Hb level were independent determinants of the PhA, after adjusting for age and BMI in men. The present study provides evidence showing that the PhA may be useful as a marker for sarcopenia, malnutrition, and cachexia in hospitalized patients with CVD.

The results of the present study show an inverse relationship between age and the PhA which is compatible with a previous study in healthy subjects [28]. The decrease in PhA values correlating with increasing age could be an indicator of a reduction in skeletal muscle mass and general health in the elderly [29,30]. In fact, the PhA correlated positively with SMI and anterior thigh MTH in both sexes in the present study. The PhA has been reported to correlate with various functional indicators [31] as well as nutritional status, frailty, and sarcopenia [32,33,34]. In our study using 412 hospitalized patients with CVD, the PhA correlated positively with hand grip strength, knee extension strength, and SPPB score in both sexes. Multivariate regression analysis also showed that hand grip strength, BNP, and Hb were independent determinants of the PhA, after adjusting for age and BMI in men. It is well known that sarcopenia is highly prevalent among elderly patients with CVD [5,7,8] and is associated with all-cause and greater CVD mortality [35]. Recently, Kamiya et al. [6] reported that the overall sarcopenia prevalence rate was 29.7% in inpatients with CVD, and patients with sarcopenia showed a higher risk of all-cause mortality compared with patients without sarcopenia. The overall prevalence of sarcopenia in the present study was 31.9%, which is similar to that of a previous study in Japanese patients with CVD [6]. Furthermore, in our study, patients with sarcopenia had higher BNP levels and lower eGFRs compared with non-sarcopenic patients. Thus, it is compatible with previous reports that sarcopenia was more common in patients with HF or CKD. However, such patients, especially those with HF, are typically overhydrated and often have other common conditions that might cause errors in BIA [28,36]. However, among BIA variables, the PhA was reported to be less influenced by overhydration, while being a good indicator of clinical outcome [37,38]. The PhA cut-off obtained from the ROC curve in patients with sarcopenia was 4.55° for males and 4.25° for females in our study. Similarly, Killic et al. [39] reported an optimal PhA cutoff of 4.55° to detect sarcopenia in 263 community-dwelling and hospitalized older adults (> 65 years). Colin-Ramirez et al. [40] showed better survival of patients in the highest PhA quartiles and shorter survival as the PhA decreases in patients with HF. They divided patients into four groups according to quartiles: (1) PhA < 4.2°, (2) PhA 4.2–4.9°, (3) PhA 5.0–5.6°, and (4) PhA > 5.7°). They found that PhA < 4.2° was an independent predictor of mortality (relative risk 3.08, 95% CI: 1.06–8.99) in comparison with PhA > 5.7°. Furthermore, Castillo Martínez et al. [41] studied 243 patients with HF (140 with reduced ejection fraction (HFrEF) and 103 with preserved ejection fraction (HFpEF) and compared them according to New York Heart Association (NYHA) functional class. In both groups, the PhA was significantly lower in patients with NYHA classes III–IV than with NYHA classes I–II (4.8° versus 5.8° in men and 4.2° versus 4.9° in women). Thus, it is possible that in HF, besides its relevance as a parameter of fluid balance because of changes in cellular membrane integrity, the PhA has an important and independent role as a marker of HF severity and prognosis.

It is well known that nutritional risk and malnutrition are predictors of survival in patients with CVD, and they increase the risk of complications and mortality [6,9]. The European Society of Clinical Nutrition and Metabolism (ESPEN) consensus statement [42] recommends that subjects at risk of malnutrition be identified by validated screening tools, and they advocate two options for the diagnosis of malnutrition: BMI to characterize malnutrition and the combined finding of unintentional weight loss and either reduced BMI or a low fat-free mass index (FFMI), or both. In our study, BMI correlated positively with SMI, PhA, and anterior thigh MTH. Furthermore, various nutritional parameters have been used to assess nutritional status, such as serum Alb level. CONUT is a nutritional evaluation score [27] that is calculated from the serum Alb level, the TChol level, and the total lymphocyte count which are easily obtained from a blood examination. CONUT was first proposed as a comprehensive scoring system for assessing the nutritional and immune status of a patient and was demonstrated to correlate with the length of hospitalization [43]. In addition to its usefulness for assessing nutrition, CONUT has been reported to be a prognostic factor for patients with chronic diseases, such as end-stage liver disease [44] and chronic HF [45]. The present study provided evidence that PhA was significantly associated with CONUT score, Hb, and Alb level in men. Thus, it is likely that the PhA can be useful as a marker for malnutrition. A recent systematic review indicates that PhA cannot independently identify malnutrition in disease based on current body of research in patients with four disease states (liver, hospitalization, oncology and renal) [46]. The reasons of the discrepancies remain unclear, but the present study showed the first evidence for clinical usefulness of PhA in patients with CAD. However, PhA and SMI did not significantly correlate with CONUT score in females but in males. Schalk et al. [47] have reported that the association between serum albumin and grip strength was stronger in males than in females. Thus, skeletal muscle function may be more affected by nutritional states in males than in females, but the further studies are needed to clarify it.

Chronic HF is associated with loss of skeletal muscle mass and body fat that progress to cardiac cachexia, a common manifestation in patients with severe HF [48]. The prevalence of cardiac cachexia has been estimated to be 10% in the current HF population [49]. However, many definitions of cachexia have been published [25,50,51]. Among them, the criteria of Fearon et al. [51], based on a generic definition proposed earlier by Evans et al. [25], has been generally used. In the present study, a diagnosis of cachexia was simply determined to meet BMI < 20 kg/m^2^, FFMI (≤17.4 kg/m^2^ for males and ≤15 kg/m^2^ for females) [26], and at least two additional biochemical criteria (Hb level < 12 g/dL, CRP level < 5 mg/dL, Alb level < 3.2 g/dL). We identified cachexia in 11.5% of males and 14.1% of females based on these cachexia criteria. The PhA cut-off obtained from the ROC curve in males with cachexia was 4.15°, while the ROC curve for females had a low AUC and sensitivity. Thus, it seems difficult to select cachexia in females by using PhA. The reasons for sex differences remains unclear, but it may be partly due to small sample size of females. Therefore, further studies using a large number of patients are required to clarify this possibility.

Our study has some limitations. First, it was a single-center observational study with a small number of patients with CVD, especially females, admitted to our hospital. In addition, the pathological conditions of enrolled patients were very different (i.e., post-operative cardiovascular surgery patients and patients admitted to the hospital for an emergency). Therefore, the nutrition states might be different. The further analysis by stratifying the patients with specific CVD is required. Furthermore, the external validity of our results for patients with CVD in the community is unclear. Secondly, the present study used cachexia criteria of BMI < 20 kg/m^2^ and FFMI (≤17.4 kg/m^2^ for males and ≤15 kg/m^2^ for females) [26] and at least two additional biochemical items (Hb level < 12 g/dL, CRP level < 5 mg/dL, Alb level < 3.2 g/dL) which did not satisfy with the criteria proposed by Evans et al. [25]. Therefore, further studies using the full criteria satisfied with the proposal of Evans et al. [25] are needed. Thirdly, PhA is determined using three main factors: age, gender, and BMI. With aging, PhA tends to decrease because of loss of skeletal muscle that translates into a reduced body reactance; on the other hand, resistance may increase due to the reduction of water content concomitantly with an increase in fat mass [52]. In what concerns gender, PhA is higher in men than women due to the greater muscle mass compartment. As for BMI, it has been observed that PhA may increase in higher BMIs because of the higher number of cells (adipocytes or muscle cells) [53]. Thus, the PhA reference values, standardized for age, gender, and BMI are mandatory for PhA analysis [54]. Therefore, PhA may be encouraged to calculate the standardized phase angle based on established population reference values stratified by a combination of age, sex, BMI, or ethnicity. Lastly, it has been reported that the lean tissue imaging is a new era for nutritional assessment [55], and reduced lean mass (LM), the best surrogate for skeletal muscle mass, is independently associated with muscle strength, ultimately leading to reduced quality of life and worse prognosis [8]. It is interesting to investigate the relations between LMI and the BIA parameters including PhA in patents with CVD.

## 5. Conclusions

The present study provided evidence showing that the PhA may be useful as a marker for sarcopenia, malnutrition, and cachexia in hospitalized patients with CVD. Thus, BIA-derived PhA may become a useful surrogate for nutritional evaluation and could help clinical practitioners aiming to diagnose malnutrition in patients with CVD.

## Figures and Tables

**Figure 1 jcm-09-02554-f001:**
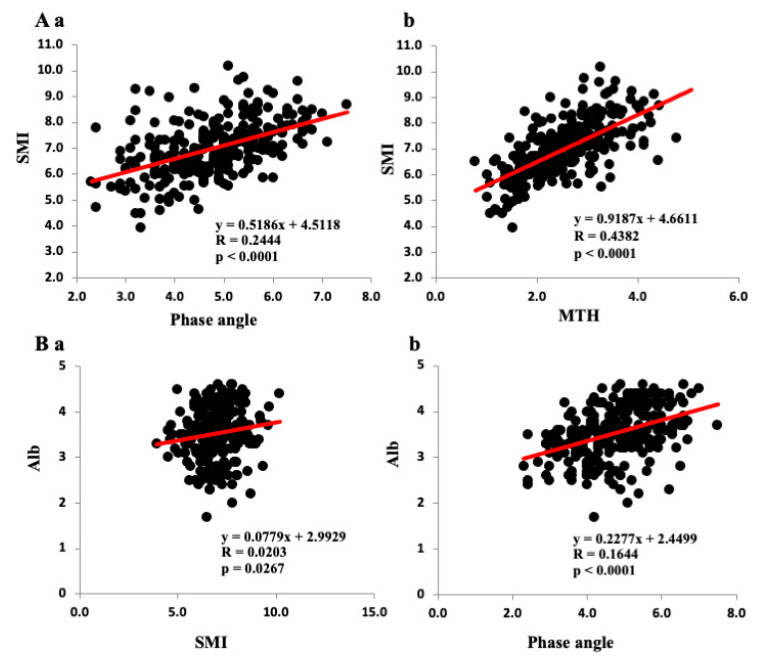
Relationships among the phase angle, SMI, and clinical data. (**A**): Relationships among the SMI, phase angle (**Aa**) and anterior thigh muscle thickness MTH (**Ab**) in all patients. (**B**): Relationships among albumin (Alb) level, the SMI (**Ba**), and the phase angle (**Bb**).

**Figure 2 jcm-09-02554-f002:**
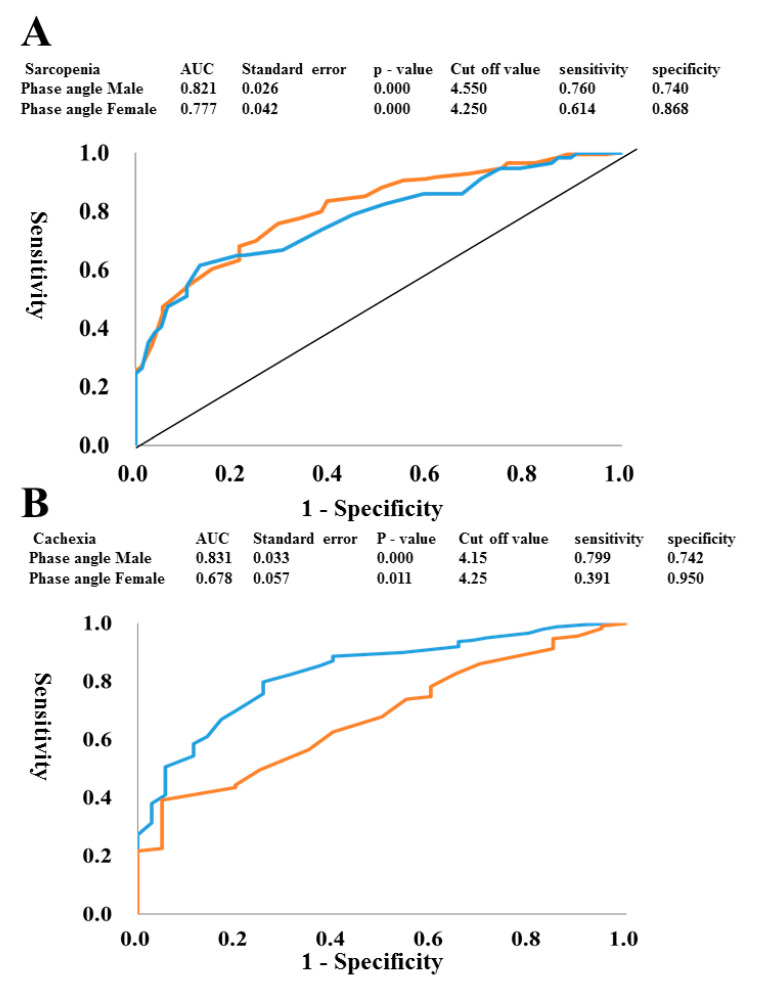
ROC curves to identify the optimal phase angle cut-off for detecting sarcopenia or cachexia. To generate the ROC curves shown, different phase angle cut-offs were used to predict sarcopenia (**A**) and cachexia (**B**) with true positives plotted on the vertical axis (sensitivity) and false positives (1-specificity) plotted on the horizontal axis in both males and females. (**A**) red (male) blue (female) (**B**) blue (male) red (female).

**Table 1 jcm-09-02554-t001:** Patient physical characteristics and clinical data.

	Males (*n* = 277)	Females (*n* = 135)
Specific diseases, *n* (%)		
Surgical disease		
CABG	52 (18.8%)	7 (5.6%)
Valve surgery	36 (13.0%)	26 (19.3%)
Aortic surgery	35 (12.6%)	15 (6.4%)
ASO	6 (2.1%)	2 (1.5%)
TAVI	8 (3.0%)	18 (13.3%)
Others	20 (7.2%)	3 (2.2%)
Internal disease		
CHF	70 (25.3%)	47 (34.8%)
IHD	75 (27.1%)	20 (14.8%)
Others	1 (0.3%)	0 (0%)
Risk factor		
HT	177 (64.0%)	94 (69.6%)
HL	155 (56.0%)	54 (40.0%)
DM	124 (44.8%)	30 (22.2%)
CKD	53 (19.1%)	17 (13.0%)
Age, years	67.7 (12.5) ***	74.6 (11.4)
Standing height, cm	164.7 (6.4) ***	149.0 (7.5)
Body weight, kg	64.8 (12.5) ***	49.0 (11.5)
BMI, kg/m^2^	23.6 (3.8) ***	22.1 (4.3)
% body fat	27.7 (7.6) ***	33.3 (10.1)
Body fat mass, kg	18.3 (7.4)	17.3 (8.1)
Systolic BP, mmHg	113.9 (16.6)	118.3 (18.9)
Diastolic BP, mmHg	64.5 (11.9)	66.0 (11.6)
BNP, pg/mL	464.0 (639.4)	455.2 (550.1)
eGFR, ml/min/1.73 m^2^	60.9 (27.1)	61.3 (28.1)
Hb, g/dL	12.4 (2.2) ***	11.5 (1.5)
Alb, g/dL	3.5 (0.6)	3.6 (0.6)
TChol, mg/dL	160.9 (40.6) ***	184.8 (47.1)
CRP, mg/dL	2.00 (3.91) **	1.38 (3.09)
Hand grip, kgf	28.9 (8.6) ***	16.1 (4.5)
Knee extension, kgf	27.2 (11.7) ***	14.5 (6.0)
CONUT score	3.21 (2.62)	2.81 (2.36)
LMI, kg/m^2^	16.7 (2.0) ***	14.4 (1.5)
SMI, kg/m^2^	6.99 (1.06) ***	5.20 (0.96)
Phase angle, °	4.79 (1.02) ***	3.99 (0.89)
MTH, cm	2.55 (0.78) ***	2.04 (0.66)
SPPB (total)	10.5 (2.2) ***	8.9 (3.1)
Balance (score)	3.68 (0.83) **	3.31 (1.17)
Gait speed, s/5 m	1.04 (0.38) ***	0.91 (0.90)
Chair stand, s/5 reps	11.7 (5.1) ***	14.4 (7.6)

*** *p* < 0.001, male versus female. ** *p* < 0.01, male versus female. Data are shown as mean ± SD or number (%) of patients. CABG, coronary artery bypass grafting; ASO, arteriosclerosis obliterans; TAVI, transcatheter aortic valve implantation; CHF, congestive heart failure; IHD, ischemic heart disease; HT, hypertension; HL, hyperlipidemia; DM, diabetes mellitus; CKD, chronic kidney disease; BMI, body mass index; BP, blood pressure; eGFR, estimated glomerular filtration rate; BNP, brain natriuretic peptide; Hb, hemoglobin; Alb, albumin; TChol, total cholesterol; CRP, C-reactive protein; kgf, kilogram-force; CONUT score, controlling nutritional status score; LMI, lean mass index; SMI, skeletal muscle mass index; MTH, anterior thigh muscle thickness; SPPB, short physical performance battery.

**Table 2 jcm-09-02554-t002:** Correlations between the SMI, phase angle, anterior thigh muscle thickness, and clinical variables.

Variable	Males (*n* = 277) SMI *r*-Value (*p*-Value)	Phase Angle *r*-Value (*p*-Value)	MTH *r*-Value (*p*-Value)	Females (*n* = 135) SMI *r*-Value (*p*-Value)	Phase Angle *r*-Value (*p*-Value)	MTH *r*-Value (*p*-Value)
Age	−0.3387 ****	−0.3120 ****	−0.3681 ****	−0.4590 ****	−0.3230 ****	−0.3900 ****
BMI	0.7290 ****	0.3310 ****	0.5768 ****	0.6076 ****	0.3115 ***	0.4840 ****
Hand grip	0.5948 ****	0.6713 ****	0.5542 ****	0.5551 ****	0.5989 ****	0.4726 ****
Knee extension MVC	0.5523 ****	0.6053 ****	0.5968 ****	0.4818 ****	0.5260 ****	0.4344 ****
CONUT	−0.2088 *	−0.4190 ****	−0.2753 ***	−0.0715	−0.2198	−0.2687 *
Hb	0.2385 ***	0.4668 ****	0.4067 ****	0.1343	0.2066 *	0.1071
Alb	0.1424 *	0.4055 ****	0.2467 ****	0.1509	0.3115 ***	0.4840 **
eGFR	0.1679 **	0.4183 ****	0.2360 ***	0.0671	0.2549 **	0.1167
BNP	−0.2397 **	−0.3527 ****	−0.3858 ****	−0.1992	−0.3268 **	−0.1849
SMI	–	0.4944 ****	0.6620 ****	–	0.4438 ****	0.5958 ***
Phase angle	0.4944 ****	–	0.6683 ****	0.4438 ****	–	0.5942 ****
MTH	0.6620 ****	0.6683 ****	–	0.5958 ****	0.5942 ****	–
SPPB (total)	0.2088 *	0.4293 ****	0.3197 ****	0.3826 ****	0.5325 ****	0.3237 ***
Balance	0.1457 *	0.2553 ****	0.0910	0.4017 ****	0.3977 ****	0.2559 **
Gait speed	0.2402 ***	0.4112 ****	0.3070 ****	0.1404	0.0764	0.2932 **
Chair stand	−0.2403 ***	−0.4666 ****	−0.3840 ****	−0.2773 ***	−0.3652 ****	−0.3376 ***

**** *p* < 0.0001, *** *p* < 0.001, ** *p* < 0.01, * *p* < 0.05. BMI, body mass index; CONUT score, controlling nutritional status score; Hb, hemoglobin; Alb, albumin; eGFR, estimated glomerular filtration rate; BNP, brain natriuretic peptide; SMI, skeletal muscle mass index; MTH, anterior thigh muscle thickness; SPPB, short physical performance battery.

**Table 3 jcm-09-02554-t003:** Multivariate linear regression analysis of the phase angle/SMI and various parameters (males/females).

	Dependent Variable: Phase Angle/SMI		
	Model 1	Model 2	Model 3
Independent Variable	β-Value (*p*) Males/Females	β-Value (*p*) Males/Females	β-Value (*p*) Males/Females
Hand grip	0.378 (0.000)/0.453 (0.001) 0.554 (0.000)/0.593 (0.000)	0.328 (0.000)/0.370 (0.006) 0.400 (0.000)/0.373 (0.003)	0.316 (0.000)/0.320 (0.020) 0.376 (0.000)/0.266 (0.024)
Knee extension (weight)	0.049(0.539)/0.057 (0.630) −0.097 (0.287)/−0.169 (0.202)	0.105 (0.198)/0.136 (0.273) 0.071 (0.311)/0.039 (0.735)	0.104 (0.200)/0.144 (0.242) 0.077 (0.263)/0.056 (0.599)
BNP log	−0.239 (0.004)/−0.163 (0.141) −0.311 (0.001)/−0.070 (0.565)	−0.206 (0.011)/−0.137 (0.206) −0.203 (0.004)/−0.002 (0.983)	−0.206 (0.012)/−0.152 (0.161) −0.198 (0.005)/−0.034 (0.712)
eGFR	0.052 (0.489)/0.022 (0.836) −0.100 (0.252)/−0.129 (0.269)	0.065 (0.374)/0.108 (0.340) −0.054 (0.404)/0.099 (0.345)	0.064 (0.377)/0.092 (0.412) −0.057 (0.368)/0.065 (0.715)
CRP log	0.176 (0.035)/0.000 (0.997) 0.057 (0.618)/0.233 (0.076)	0.138 (0.093)/−0.060 (0.614) −0.079 (0.273)/0.076 (0.493)	0.136 (0.099)/−0.091 (0.448) −0.088 (0.214)/0.008 (0.937)
Hb	0.291 (0.000)/0.230 (0.038) 0.128 (0.153)/−0.028 (0.815)	0.239 (0.003)/0.207 (0.058) −0.027 (0.694)/−0.091 (0.362)	0.227 (0.005)/0.209 (0.054) −0.048 (0.483)/−0.087 (0.342)
Alb log	0.160 (0.065)/−0.062 (0.607) 0.038 (0.702)/−0.109 (0.416)	0.134 (0.115)/−0.059 (0.619) −0.053 (0.481)/−0.100 (0.364)	0.140 (0.101)/−0.066 (0.576) −0.047 (0.520)/−0.114 (0.256)

Model 1, unadjusted; Model 2, adjusted for BMI; Model 3, adjusted for BMI and age.

**Table 4 jcm-09-02554-t004:** Comparison of various parameters between patients with and without sarcopenia.

Variable	Males (*n* = 234) Sarcopenia *n* = 74	Others *n* = 160	Females (*n* = 114) Sarcopenia *n* = 37	Others *n* = 77
Age, years	74.9 (9.2) ****	64.9 (11.2)	79.4 (8.1) ****	71.7 (12.1)
BMI, kg/m^2^	21.1 (2.9) ****	24.7 (3.5)	21.4 (3.8)	22.8 (4.5)
% body fat	27.0 (8.3)	27.7 (7.1)	34.5 (10.6)	32.9 (9.8)
BNP, pg/mL	683 (815) ***	346 (489)	613 (640)	371 (405)
eGFR, pg/mL	51.5 (28.2) ***	66.0 (24.7)	59.0 (33.4)	62.2 (24.8)
Hand grip, kgf	21.1 (4.2) ****	32.8 (7.1)	12.8 (3.3) ****	18.2 (3.8)
Knee extension, kgf	18.1 (6.1) ****	31.3 (10.8)	11.0 (4.4) ****	16.5 (5.6)
CONUT score	4.3 (2.6) ****	2.5 (2.3)	3.2 (2.4)	2.3 (2.0)
Hb, g/dL	11.7 (1.8) ****	12.8 (2.2)	11.1 (1.4) *	11.8 (1.6)
Alb, g/dL	3.4 (0.5) ***	3.6 (0.6)	3.4 (0.6) *	3.7 (0.6)
SMI, kg/m^2^	5.92 (0.57) ****	7.54 (0.84)	4.58 (0.61) ****	5.64 (0.85)
LMI, kg/m^2^	15.1 (1.5) ****	17.6 (2.0)	13.7 (1.1) ****	14.9 (1.4)
Phase angle	4.05 (0.79) ****	5.19 (0.87)	3.62 (0.69) ****	4.30 (0.88)
MTH, cm	1.96 (0.64) ****	2.91 (0.67)	1.75 (0.57) ****	2.28 (0.64)
SPPB (total)	8.9 (2.3) ****	11.1 (1.6)	6.3 (2.6) ****	10.1 (2.2)
Balance (score)	3.54 (0.97) **	3.78 (0.67)	2.64 (1.34) ****	3.69 (0.78)
Gait speed, s/5 m	0.83 (0.23) ****	1.14 (0.39)	0.56 (0.13) ****	1.08 (1.06)
Chair stand, s/5 reps	15.4 (6.7) ****	10.5 (3.6)	18.8 (8.4) ****	12.6 (4.6)

**** *p* < 0.0001, *** *p* < 0.001, ** *p* < 0.01, * *p* < 0.05 sarcopenia versus others (males, females). Data are shown as the mean ± SD. Number, number of patients examined. BMI, body mass index; CONUT score, controlling nutritional status score; Hb, hemoglobin; Alb, albumin; eGFR, estimated glomerular filtration rate; kgf, kilogram-force; BNP, brain natriuretic peptide; SMI, skeletal muscle mass index; LMI, lean mass index; MTH, anterior thigh muscle thickness; SPPB, short physical performance battery.

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
