# Peer review of "Phase Angle as an Indicator of Sarcopenia, Malnutrition, and Cachexia in Inpatients with Cardiovascular Diseases"

_jcm, 2020, doi:10.3390/jcm9082554_

Round 1
Reviewer 1 Report
The manuscript entitled “Phase Angle as an Indicator of Sarcopenia, Malnutrition, and Cachexia in Inpatients with Cardiovascular Diseases” assessed the usefulness of the PhA as a marker of sarcopenia, malnutrition, and cachexia in hospitalized patients with cardiovascular disease (CVD).
My comments are as follows:
- The study evaluated the body composition with bioelectrical impedance analysis and nutritional status in hospitalized patients with cardiovascular disease (CVD). However, CVD is an umbrella term that includes a wide variety of conditions in which heterogeneous susceptibility to malnutrition may exist. It is advisable that the authors should perform the analysis by stratifying the patients by specific CVD.
- Introduction: The introduction section mentions the impact of increased nutritional risk, malnutrition, and sarcopenia on chronic heart failure (HF) only. Similar issue is present in the Discussion section. Please expand the content by discussing their impact on other cardiovascular diseases and CVD in general.
- Methods: “Diagnosis of cachexia was determined to meet BMI < 20 kg/m2 and at least two following biochemical criteria (Hb <12 g/dl, CRP <5 mg/dl, Alb <3.2 g/dl).” Please provide the reference. The definition of cachexia used in the study deviates from the diagnostic criteria for cachexia in adults developed by the cachexia consensus panel (Evans et al., Clin Nutr. 2008 Dec;27(6):793-9), which requires the presence of weight loss of at least 5% in 12 months or less in the presence of underlying illness plus at least three of the five criteria (decreased muscle strength, fatigue, anorexia, low fat-free mass index, and abnormal biochemistry). Using a non-standard definition of cachexia may affect the validity of the study results.
- Methods: PhA has been reported to be confounded by level of fluid status, body composition, and physical activity. To account for confounding, it is encouraged to calculate the standardized phase angle based on established population reference values stratified by a combination of age, sex, BMI, or ethnicity.
- Methods: Details of obtaining the optimal cut-off of PhA should be reported.
- Results, Table 2: It is noteworthy that unlike males, PhA and SMI did not significantly correlate with CONUT score in females. Please comment and provide potential explanations.
- Results, Figure 1: The strength of correlations between PhA and SMI or Alb were rather weak despite statistical significance, making their clinical relevance questionable.
- Results, Figure 2: The usefulness of PhA for discriminating sarcopenia and cachexia appears to be suboptimal in females considering the c-statistic.
- Discussion: A recent systematic review indicates that PhA cannot independently identify malnutrition in disease based on current body of research. In addition, the measurement of PhA may be influenced by the operator (Clin Nutr ESPEN. 2019 Feb;29:1-14). Please comment.
- Another limitation of using a PhA cut-off value is dichotomization, which leads to several problems such as loss of statistical power by discarding data, increasing the risk of false positivity, underestimating the extent of variation between groups, not accounting for the non-linear relationship.
- Discussion, Page 11: “The present study provides evidence showing that the PhA can be highly useful as a marker for sarcopenia, malnutrition and cachexia in hospitalized patients with CVD.” The conclusion should be toned down in view of abovementioned limitations.
Author Response
Reply to Reviewer #1
We greatly appreciate your careful attention to our manuscript and especially your excellent suggestions for improving the clarity and correctness of the message. We have corrected the paper as per your suggestions, and consider the revised manuscript much improved.
Reviewer #1:
The manuscript entitled “Phase Angle as an Indicator of Sarcopenia, Malnutrition, and Cachexia in Inpatients with Cardiovascular Diseases” assessed the usefulness of the PhA as a marker of sarcopenia, malnutrition, and cachexia in hospitalized patients with cardiovascular disease (CVD).
My comments are as follows: The study evaluated the body composition with bioelectrical impedance analysis and nutritional status in hospitalized patients with cardiovascular disease (CVD). However, CVD is an umbrella term that includes a wide variety of conditions in which heterogeneous susceptibility to malnutrition may exist. It is advisable that the authors should perform the analysis by stratifying the patients by specific CVD.
#) Answer: You are absolutely right. However, as mentioned in methods, our study included a total of 412 patients who underwent cardiac rehabilitation on admission due to CVD, and therefore contained a variety of disease. Your comments are absolutely right, but it is difficult to stratify the patients by specific CVD. Therefore, we mentioned it in discussion.
p 12 line 389~.
In addition, the pathological condition of enrolled patients was very different (i.e. post-operative cardiovascular surgery patients, and patients admitted to the hospital for an emergency). Therefore, the nutrition status might be different. The further analysis by stratifying the patients with specific CVD is required.
Introduction: The introduction section mentions the impact of increased nutritional risk, malnutrition, and sarcopenia on chronic heart failure (HF) only. Similar issue is present in the Discussion section. Please expand the content by discussing their impact on other cardiovascular diseases and CVD in general.
#) Answer: We expanded the content by discussing their impact on other cardiovascular diseases, and added the two papers.
P11 line 325~
and is associated with all-cause and greater CVD mortality [34].
p11 line 350~
It is well known that nutritional risk, and malnutrition are predictors of survival in patients with CVD, and they increase the risk of complications and mortality [6,9].
We cited two papers.
Chin SO, Rhee SY, Chon S, Hwang YC, Jeong IK, Oh S, Ahn KJ, Chung HY, Woo JT, Kim SW, Kim JW, Kim YS, Ahn HY. Sarcopenia is independently associated with cardiovascular disease in older Korean adults: the Korea National Health and Nutrition Examination Survey (KNHANES) from 2009. PLoS One. 2013;8(3):e60119. doi: 10.1371/journal.pone.0060119. Epub 2013 Mar 22.
Boban M, Bulj N, Kolačević Zeljković M, Radeljić V, Krcmar T, Trbusic M, Delić-Brkljačić D, Alebic T, Vcev A. Nutritional Considerations of Cardiovascular Diseases and Treatments. Nutr Metab Insights. 2019 Mar 22;12:1178638819833705. doi: 10.1177/1178638819833705. eCollection 2019.
Methods: “Diagnosis of cachexia was determined to meet BMI < 20 kg/m2 and at least two following biochemical criteria (Hb <12 g/dl, CRP <5 mg/dl, Alb <3.2 g/dl).” Please provide the reference. The definition of cachexia used in the study deviates from the diagnostic criteria for cachexia in adults developed by the cachexia consensus panel (Evans et al., Clin Nutr. 2008 Dec;27(6):793-9), which requires the presence of weight loss of at least 5% in 12 months or less in the presence of underlying illness plus at least three of the five criteria (decreased muscle strength, fatigue, anorexia, low fat-free mass index, and abnormal biochemistry. Using a non-standard definition of cachexia may affect the validity of the study results.
#) Answer: We are sorry that the criteria of FFMI was lacking. We agree with your opinion. As your suggestion, Evans et al. have showed the criteria as follows. Cachexia has been defined as a loss of lean tissue mass, involving a weight loss greater than 5% of body weight in 12 months or less in the presence of chronic illness or as a body mass index (BMI) lower than 20 kg/m2. In addition, usually three of the following five criteria are required: decreased muscle strength, fatigue, anorexia, low fat-free mass index, increase of inflammation markers such as C-reactive protein or interleukin (IL)-6 as well as anemia or low serum albumin (CRP>5.0 mg/l, IL-6>4.0 pg/ml, Hb<12 g/dl, Alb 3.2 g/dl). In the present study, all the subjects had CVD. But unfortunately, we have not checked the body loss, and we have chosen BMI criteria. And, we also selected the FFMI criteria (≦ 17.4 kg/m2 for males and ≦ 15 kg/m2 for females)[26] and at least two following biochemical criteria (Hb < 12 g/dl, CRP < 5 mg/dl, Alb < 3.2 g/dl). Therefore, the criteria of cachexia in the present study did not satisfy with the criteria proposed by Evans et al. (2008). We mentioned it in methods and discussion (limitation).
p4 line 148~
Cachexia has been defined by Evans et al. [25] as a loss of lean tissue mass, involving a weight loss greater than 5% of body weight in 12 months or less in the presence of chronic illness or as BMI lower than 20 kg/m2. In addition, usually three of the following five criteria are required: decreased muscle strength, fatigue, anorexia, low fat-free mass index (FFMI), increase of inflammation markers such as C-reactive protein or interleukin (IL)-6 as well as anemia or low serum albumin (CRP > 5.0 mg/l, IL-6 > 4.0 pg/ml, Hb < 12 g/dl, Alb < 3.2 g/dl). In the present study, cachexia was determined to meet BMI < 20 kg/m2 and FFMI (≦ 17.4 kg/m2 for males and ≦ 15 kg/m2 for females)[26] and at least two following biochemical criteria (Hb < 12 g/dl, CRP < 5 mg/dl, Alb < 3.2 g/dl).
p12 line 394~
Secondly, the present study used cachexia criteria of BMI < 20 kg/m2 and FFMI (≦ 17.4 kg/m2 for males and ≦ 15 kg/m2 for females)[26] and at least two additional biochemical items (Hb level < 12 g / dl, CRP level < 5 mg / dl, Alb level < 3.2 g / dl), which did not satisfy with the criteria proposed by Evans et al. [25]. Therefore, the further studies using the full criteria satisfied with the proposal of Evans et al. [25] are needed.
Methods: PhA has been reported to be confounded by level of fluid status, body composition, and physical activity. To account for confounding, it is encouraged to calculate the standardized phase angle based on established population reference values stratified by a combination of age, sex, BMI, or ethnicity.
#) Answer: Thank you very much for your comments. We commented it in discussion.
p12 line 398~
Thirdly, PhA is determined with 3 main factors: age, gender and BMI. With aging, PhA tends to decrease because of loss of skeletal muscle that translates into a reduced body reactance; on the other hand, resistance may increase due to a reduction on water content concomitantly with an increase in fat mass [51]. In what concerns gender, PhA is higher in men than women due to a greater muscle mass compartment. As for BMI, it has been observed that PhA may increase in higher BMIs because of the higher number of cells (adipocytes or muscle cells) [52]. Thus, the PhA reference values, standardized for age, gender and BMI are mandatory for PhA analysis [53]. Therefore, PhA may be encouraged to calculate the standardized phase angle based on established population reference values stratified by a combination of age, sex, BMI, or ethnicity.
We cited the following 3 papers.
Toso S, Piccoli A, Gusella M, Menon D, Bononi A, Crepaldi G, Ferrazzi E. Altered tissue electric properties in lung cancer patients as detected by bioelectric impedance vector analysis. Nutrition. 2000;16:120-124.
Paiva, IS., Borges, RL., Halpern-Silveira, D., Assuncao, FMC., Barros, JDA., Gonzalez, CM. Standardized phase angle from bioelectrical impedance analysis as prognostic factor for survival in patients with cancer. Support Care Cancer (2011) 19, 187-192.
Gupta, D., Lis, GC., Dahlk, LS., Vashi, GP., Grutsch, FJ., Lammersfeld, AC. Bioelectrical impedance phase angle as a prognostic indicator in advanced pancreatic cancer. British Journal of Nutrition (2004) 92, 957-962.
Methods: Details of obtaining the optimal cut-off of PhA should be reported.
#) Answer: I added the method to obtain the optimal cut-off of PhA in methods.
p5 line 178~
Receiver operating characteristic (ROC) curves were plotted to identify an optimal PhA cut-off for detecting sarcopenia or cachexia. With or without sarcopenia or cachexia as dependent factors, the sensitivity, specificity and false positive rate (1-specificity) of the phase angle were calculated to obtain the ROC curve. At this time, the Youden index (sensitivity+specificity-1) was calculated from the obtained sensitivity and specificity, and the point at the maximal value was taken as the optimum cutoff value.
Results, Table 2: It is noteworthy that unlike males, PhA and SMI did not significantly correlate with CONUT score in females. Please comment and provide potential explanations.
#) Answer: In the present analysis, PhA and SMI did not significantly correlate with CONUT score in females, but in males. The reasons of the difference between males and females remain unclear. However, Schalk et al. (2005) have reported that the association between serum albumin and grip strength was stronger in men than in women. Thus, skeletal muscle function may be more affected by nutritional states in males than in females, but the further studies are needed to clarify it.
p12 line 369~
However, PhA and SMI did not significantly correlate with CONUT score in females, but in males. Schalk et al. [46] have reported that the association between serum albumin and grip strength was stronger in men than in women. Thus, skeletal muscle function may be more affected by nutritional states in males than in females, but the further studies are needed to clarify it.
Schalk BW, Deeg DJ, Penninx BW, Bouter LM, Visser M. Serum albumin and muscle strength: A longitudinal study in older men and women. J Am Geriatr Soc. 2005;53:1331–1338.
Results, Figure 1: The strength of correlations between PhA and SMI or Alb were rather weak despite statistical significance, making their clinical relevance questionable.
#)Answer: Thank you very much for your suggestions. I agree with your comments.
The present study has utilized large sample sizes, and therefore can result in associations that are small in magnitude with highly statistically significant p-values. But, we cannot rule out another possibility of bias. Until now, many papers have reported the relationships between SMI and Alb in patients with various diseases. Also, in this study, there were significant correlations between SMI and Alb. Therefore, it is likely that PhA has a significant correlation with Alb, and similarly SMI in patients with CVD. We did not mention about it.
Results, Figure 2: The usefulness of PhA for discriminating sarcopenia and cachexia appears to be suboptimal in females considering the c-statistic.
#) Answer: You are absolutely right. We could not discriminate cachexia in females considering the c-statistic.
The sensitivity and specificity of PhA for discriminating cachexia were 79.9% and 74.2% in males, respectively. In contrast, the AUC and sensitivity of AUC curve were low in females. Therefore, it seems difficult to select cachexia in females by using phase angle.
p9 line 295
In contrast, the sensitivity of AUC curve in females was low (39.1%).
p12 line 382~
The PhA cut-off obtained from the ROC curve in males with cachexia was 4.15°, while the ROC curve for females had a low AUC and sensitivity. Thus, it seems difficult to select cachexia in females by using phase angle. The reasons of sex differences remains unclear, but it may be partly due to ta small sample size of females. Therefore, the further studies using a large number of patients are required to clarify this possibility.
Discussion: A recent systematic review indicates that PhA cannot independently identify malnutrition in disease based on current body of research. In addition, the measurement of PhA may be influenced by the operator (Clin Nutr ESPEN. 2019 Feb;29:1-14). Please comment.
#) Answer: Thank you very much for your suggestion. As your suggestion, a recent systematic review indicates that PhA cannot independently identify malnutrition in disease based on current body of research in patients with four disease states (liver, hospitalization, oncology and renal). However, the present study showed the first study for clinical usefulness of PhA in patients with CAD. However, several limitations exist in this study, and we described it in discussion.
Rinaldi S, Gilliland J, O'Connor C, Chesworth B, Madill J. Is phase angle an appropriate indicator of malnutrition in different disease states? A systematic review. Clin Nutr ESPEN. 2019 Feb;29:1-14.
page 12 line 365~
Thus, it is likely that the PhA can be useful as a marker for malnutrition. A recent systematic review indicates that PhA cannot independently identify malnutrition in disease based on current body of research in patients with four disease states (liver, hospitalization, oncology and renal) [45]. The reasons of the discrepancies remain unclear, but the present study showed the first evidence for clinical usefulness of PhA in patients with CAD.
Another limitation of using a PhA cut-off value is dichotomization, which leads to several problems such as loss of statistical power by discarding data, increasing the risk of false positivity, underestimating the extent of variation between groups, not accounting for the non-linear relationship.
#) Answer: Thank you very much for your suggestion. We absolutely agree with your opinion.
(1) The number of patients with cachexia was significantly small, especially in females. (2) The PhA data of the females used in this study were non-normal distribution data, and the median value was 3.9. (3) While there are various biases (evaluation items) in the evaluation of the pathophysiology of cachexia, there is a possibility that one aspect of PhA may not be able to make a sufficient judgment. Thus, in the ROC model of cachexia in this study, especially for women, the median PhA was low (3.9). In addition, because the ROC curve was based on data with fewer patients in the population, it is thought that the AUC and sensitivity were low., and it is difficult to select the optimum cutoff value of the PhA in the presence or absence of cachexia in females. I added the following sentences in discussion.
P12 line 382~.
The PhA cut-off obtained from the ROC curve in males with cachexia was 4.15°, while the ROC curve for females had a low AUC and sensitivity. Thus, it seems difficult to select cachexia in females by using PhA. The reasons of sex differences remains unclear, but it may be partly due to small sample size of females. Therefore, the further studies using a large number of patients are required to clarify this possibility.
Discussion, Page 11: “The present study provides evidence showing that the PhA can be highly useful as a marker for sarcopenia, malnutrition and cachexia in hospitalized patients with CVD.” The conclusion should be toned down in view of abovementioned limitations.
#) Answer: I agree with your opinion. We changed it as follows.
The present study provides evidence showing that the PhA may be useful as a marker for sarcopenia, malnutrition and cachexia in hospitalized patients with CVD

Reviewer 2 Report
The article is of interest as the use of BIA provides crucial information on the nutritional status of individuals, yet it is extremely underused, especially in the clinical non-research setting. For such reason the authors are congratulated for providing important evidence to promote its use. They also provide cut-off for phase angle, which is quite useful.
I would only suggest minor recommendations:
- Authors describe muscle mass and lean mass as if they were the same body composition compartment. I would avoid doing so, and instead of using SMI, using LMI (lean mass index) as the index was obtained using lean mass and not skeletal muscle mass.(1)
- I would include a recent article that describes the role of lean mass in patients with CVD, particularly heart failure. (2)
- I would suggest to include more numerical data in the abstract, including p values, and reducing the non-numerical language in the abstract.
References
- Prado CM, Heymsfield SB. Lean tissue imaging: a new era for nutritional assessment and intervention. JPEN Journal of parenteral and enteral nutrition. 2014;38(8):940-53.
- Carbone S, Billingsley HE, Rodriguez-Miguelez P, Kirkman DL, Garten R, Franco RL, et al. Lean Mass Abnormalities in Heart Failure: The Role of Sarcopenia, Sarcopenic Obesity, and Cachexia. Curr Probl Cardiol. 2019:100417.
Author Response
Reply to Reviewer #2
We greatly appreciate your careful attention to our manuscript and especially your excellent suggestions for improving the clarity and correctness of the message. We have corrected the paper as per your suggestions, and consider the revised manuscript much improved.
Reviewer #2:
The article is of interest as the use of BIA provides crucial information on the nutritional status of individuals, yet it is extremely underused, especially in the clinical non-research setting. For such reason the authors are congratulated for providing important evidence to promote its use. They also provide cut-off for phase angle, which is quite useful.
I would only suggest minor recommendations:
- Authors describe muscle mass and lean mass as if they were the same body composition compartment. I would avoid doing so, and instead of using SMI, using LMI (lean mass index) as the index was obtained using lean mass and not skeletal muscle mass.(1).
#) Answer: Thank you very much for your suggestion. But, the present study used SMI, but not LMI for diagnosis of sarcopenia. Therefore, we presented the data of SMI. In table 1& 4, we also added the data of LMI.
2 .I would include a recent article that describes the role of lean mass in patients with CVD, particularly heart failure. (2)
#) Answer: Thank you very much. I added the following sentence in discussion and cited this paper.
p12 line 407~
Lastly, it has been reported that the lean tissue imaging is a new era for nutritional assessment [54], and reduced lean mass (LM), the best surrogate for skeletal muscle mass, is independently associated with muscle strength, ultimately leading to reduced quality of life and worse prognosis [8]. It is interesting to investigate the relations between LMI and the BIA parameters including PhA in patents with CVD.
- I would suggest to include more numerical data in the abstract, including p values, and reducing the non-numerical language in the abstract.
#) Answer: I included p value in this abstract.
page 1 line 22~
Both SMI and PhA correlated negatively with age (p<0.0001) and positively with grip strength, and knee extension strength (all, p < 0.0001) in both sexes. SMI correlated significantly with CONUT score (p < 0.05), Hb (p < 0.001), and Alb (p < 0.05) in males. PhA also correlated significantly with CONUT score, Hb, and Alb (all p < 0.0001) in males, and the PhA was more strongly associated with these nutritional aspects. In females, PhA was correlated with Hb (p < 0.05) and Alb (p < 0.001), but not SMI.
We cited the following two papers.
References
- Prado CM, Heymsfield SB. Lean tissue imaging: a new era for nutritional assessment and intervention. JPEN J Parenter Enteral Nutr. 2014 Nov;38(8):940-953.
- Carbone S, Billingsley HE, Rodriguez-Miguelez P, Kirkman DL, Garten R, Franco RL, Lee DC, Lavie CJ. Lean Mass Abnormalities in Heart Failure: The Role of Sarcopenia, Sarcopenic Obesity, and Cachexia. Curr Probl Cardiol. 2019 Mar 28:100417. doi: 10.1016/j.cpcardiol.2019.03.006.

Round 2
Reviewer 1 Report
The previous comments have been addressed sufficiently. The revised manuscript has been improved by expanding the discussion and limitations. I have no further comments at the current stage.